# Clinical Usability of Embryo Development Using a Combined Qualitative and Quantitative Approach in a Single Vitrified-Warmed Blastocyst Transfer: Assessment of Pre-Vitrified Blastocyst Diameter and Post-Warmed Blastocyst Re-Expansion Speed

**DOI:** 10.3390/jcm11237085

**Published:** 2022-11-29

**Authors:** Jae Kyun Park, So-Yeon Ahn, Su Hee Seok, Sol Yi Park, Soyoung Bang, Jin Hee Eum, In Pyung Kwak, Ji Won Kim, Woo Sik Lee

**Affiliations:** Department of Obstetrics and Gynecology, CHA Fertility Center Gangnam, CHA University School of Medicine, Seoul 06105, Republic of Korea

**Keywords:** single vitrified-warmed blastocyst transfers, time-lapse system, morphokinetic, diameter, re-expansion speed, blastocyst quality scores

## Abstract

Improving the safety and efficacy of assisted reproductive technology programs has been a continuous challenge. Traditionally, morphological grading has been used for embryo selection. However, only a few studies have assessed the morphokinetic variables and morphological dynamics of blastocysts. In the present study, we aimed to perform a quantitative analysis of blastocyst diameter and re-expansion speed. This in-depth morphokinetic evaluation can correlate with currently observed pregnancy outcomes. In total, 658 single vitrified-warmed blastocyst transfer cycles were performed between October 2017 and December 2021, which were divided into four groups according to the pre-vitrified blastocyst diameter. After warming, the groups were subdivided according to the blastocyst re-expansion speed. These quantitative measurements were performed using a time-lapse system. Both diameter and speed are essential in determining the blastocyst quality, while age, day of freezing, and blastocyst quality are crucial from a clinical perspective. The application of both quantitative (diameter and speed) and qualitative (blastocyst quality scores) parameters can help evaluate the clinical usability of blastocysts. This method can prove useful for embryologists in counseling their patients and determining pregnancy patient-oriented strategies.

## 1. Introduction

The goal of assisted human reproduction is the birth of a healthy child. Selecting embryos with the highest developmental potential is essential in assisted reproductive technology (ART) [1,2,3]. Understanding the dynamics of the embryo developmental process is also crucial, and embryos with the highest developmental potential are selected by culturing them until the blastocyst stage [4,5,6]. Recently, single-embryo transfer approaches have been recommended for patients to minimize multiple pregnancies [7,8,9]. The selection of good-quality embryos shortens the time to achieve ongoing pregnancy and reduces the associated costs and physical and psychological stress on the patient. Therefore, understanding the dynamics of the embryo developmental process and identifying other parameters affecting the quality of the embryos is essential [10,11,12,13].

Traditionally, morphological evaluation was essential for embryo selection at the blastocyst stage [10,14]. However, there are some limitations associated with the morphological grading systems. For example, morphological evaluation was performed based on a subjective interpretation by an embryologist, and the static “snap-shot” assessment was performed in a short time. Moreover, embryos undergo dynamic morphological changes, and the morphological evaluation may miss some significant events. To overcome these, time-lapse technology was applied to make continuous observations in stable embryo culture conditions and to enable electronic documentation of embryo behavior. The time-lapse system (TLS) can measure embryo morphokinetic parameters and division speed [2,15,16]. This approach enables the selection of embryos with high developmental potential and can be used as a non-invasive tool to predict embryo development and implantation potential [1,17,18].

Infertility specialists seek a quantitative method to evaluate blastocysts and correlate them with clinical outcomes as a tool for embryo selection [19,20,21,22,23]. The selection is highly dependent on morphological grade, and only a few studies have assessed the morphokinetic variables and morphological dynamics of blastocysts. In our previous study, we observed a correlation between the degree of blastocyst expansion and implantation potential [10]. Therefore, we assumed that blastocyst expansion speed and size are important parameters for predicting embryo quality [13,24,25]. Until now, qualitative assessment of the blastocyst stage has been performed in combination with the degree of expansion, inner cell mass, and trophectoderm grade. Unfortunately, these three independent quality scores are ambiguous, making it difficult to predict the clinical outcomes. Therefore, in this study, we employed a quantitative approach to analyze the blastocyst diameter and re-expansion speed. We hypothesized that the in-depth morphological evaluation positively correlates with current pregnancy outcomes.

## 2. Materials and Methods

### 2.1. Patients and Study Design

This single-center, retrospective cohort study reviewed 658 single vitrified-warmed blastocyst transfer (SVBT) cycles performed between October 2017 and December 2021 at the affiliated CHA Gangnam Medical Center, CHA University (Seoul, Korea). For each patient, clinical and ongoing pregnancy data were extracted from an electronic medical record database, and missing data were completed via telephonic surveys. Patients who underwent any in vitro maturation protocol, oocyte donor cycle, or pre-implantation genetic testing were excluded. We also excluded women with uterine anomalies or endometrial thickness <7 mm. All SVBT cycles were divided into four groups according to the diameter of expansion: Group I, <135 μm (*n* = 158); Group II, ≥135–≤145 μm (*n* = 197); Group III, ≥146–≤155 μm (*n* = 171); and Group IV, ≥156 μm (*n* = 132). The screening was performed by three blinded senior embryologists.

### 2.2. Endometrial Preparation

Endometrial preparation in the SVBT cycle and embryo transfer were performed in accordance with our previous study [10,26,27]. In brief, two major clinical protocol categories were applied to endometrial preparation in vitrified-warmed cycles: natural cycles and hormone replacement treatment cycles. In the natural cycle, all patients were closely monitored for evidence of dominant follicle collapse using transvaginal ultrasonography during days 10–12 of the menstrual cycle. When ovulation was observed by the appearance of a double contour along the follicle wall, with the wall almost fainting away, progesterone was supplemented with Crinone gel or Utrogestan capsules. For the hormone replacement treatment cycle, patients were administered 4 or 6 mg estradiol valerate (Progynova, Schering AG, Berlin, Germany) per day from the second and third day of the menstrual cycle. After 7–10 days, the endometrial thickness was determined using transvaginal ultrasonography, and when it reached the appropriate thickness (>8 mm) for embryo transfer, luteal support was initiated, as described above.

### 2.3. Vitrified-Warmed Blastocyst Preparation

The slush-EM grid method for preparing vitrified-warmed blastocysts was used as described in our previous study [10,27]. Briefly, blastocysts were pre-equilibrated in an equilibrium solution of 7.5% (*v*/*v*) ethylene glycol (EG), 7.5% dimethyl sulfoxide (DMSO) in HEPES medium (SAGE, Cooper Surgical, Trumbull, CT, USA), and 20% human serum albumin (HSA) at 37 °C for 2.5 min. They were then placed in a vitrification solution with 15% EG, 15% DMSO, and 0.5 M sucrose. After 20 s, blastocysts were placed on the EM-gold grid with a minimal volume (<0.1 μL) of vitrification solution and immediately plunged into Slush LN_2_ using a VitMaster. For warming, the EM-gold grid was transferred to decreasing concentrations of warming solution (0.5, 0.25, 0.125, and 0.0 M of sucrose) for 2.5 min each. Blastocysts were placed in a TLS at 37 °C, with 6% CO_2_, 5% O_2_, and 89% N_2_, and cultured overnight. All chemicals used in this study were from Sigma Chemical Co., (Saint Louis, MO, USA) unless otherwise indicated.

### 2.4. Recording and Annotation of Blastocyst Culture in the TLS

A 12-well dish was prepared and equilibrated in a CO_2_ incubator for 4 h prior to embryo culture. Each well was filled with 25 µL blastocyst culture medium (COOK, QLD, Australia) and covered with 1.4 mL mineral oil (Ovoil, Vitrolife AB, Sweden). The warmed blastocysts were then placed in microwells. The dish was placed in the dish holder inside an Embryoscope™ (Vitrolife) TLS under stable conditions of 5% O_2_ and 6% CO_2_ at 37 °C. Images were recorded automatically in nine focal planes every 7 min (15 µm intervals) and saved for evaluation at an external workstation (Embryo Viewer™, Vitrolife AB, Sweden). Analysis was performed using the Embryoscope software (EmbryoViewer software). Time-lapse recordings of 658 embryos were manually annotated at an external workstation.

### 2.5. Measurements of Blastocyst Diameter and Re-Expansion Speed

The warmed blastocysts were annotated according to the published consensus definitions and guidelines [28,29]. Beginning at the time of blastocyst re-expansion, the cross-sectional area (in µm^2^) was recorded using the Embryoscope viewer’s elliptical measurement tool by circumscribing the outer periphery of the trophectoderm excluding the zona pellucida. The blastocoel re-expansion diameter was determined by averaging the three cross-sectional diameter measurements, which were made at the beginning of blastocyst re-expansion until the point when the embryo was removed for transfer. Embryo morphology was assessed based on the images acquired from the TLS using four morphokinetic parameters as follows: the time of the start of expansion (blastocoel increases in size), time of completion of re-expansion (blastocyst occupies the whole perivitelline space), start point at the time of hatching (trophectoderm blebs out of the manipulated zona pellucida), and the migration speed of expansion (total migration length/total time of filled re-expansion appearance).

### 2.6. Blastocyst Survival, Re-Expansion, and Grading Assessment

The vitrified-warmed blastocyst survival and grading assessments were performed as described in our previous study [10,30]. In brief, embryologists assessed the degree of post-warming blastocyst survival by identifying viable cells (clear margin and homogeneous content) in comparison to dead cells (dark and granular) and defined fully surviving embryos when >50% of intact cells were observed. The degree of blastocoel re-expansion was graded into four categories: collapsed (shrinkage, if <10% of the original volume was recovered), partially re-expanded (if approximately 10–95% of expansion was achieved), fully re-expanded (if approximately 100% of expansion was achieved), and hatching/hatched (blastocyst squeezes out from its zona pellucida). Blastocyst quality scores (BQS) were determined as follows: (blastocoel expansion score) × (inner cell mass score) × (trophectodermal score). Blastocysts were divided into four groups based on their BQS: top (BQS; 45–24), good (BQS; 23–16), average (BQS; 15–7), and poor (BQS; 6–1).

### 2.7. Embryo Transfer and Identification of Outcome Parameters

A warmed blastocyst was transferred under abdominal ultrasound guidance using a soft catheter for embryo transfer 6 days post ovulation or progesterone supplementation. In all groups, serum β-human chorionic gonadotropin (β-hCG) tests were performed 2 weeks after vitrified-warmed embryo transfer. An increase in serum β-hCG level (>20 IU) was considered positive. Pregnancy and implantation were defined by ultrasound observation of the gestational sac. Ongoing pregnancy was defined by the presence of a fetal heartbeat on ultrasonography. Miscarriage was defined as a spontaneous discontinuation/clinical pregnancy.

### 2.8. Statistical Analysis

Statistical calculations were performed using SPSS ver. 25 (IBM, Armonk, NY, USA). Continuous data are expressed as the mean ± standard deviation, whereas categorical variables are expressed as percentages. Normality distribution was assessed using Levene’s test for all variables. Data were analyzed using ANOVA followed by Duncan’s multiple range test. Data of the continuous variables were analyzed using the Student’s *t*-test, and ordinary variables such as rate comparisons were analyzed using the chi-square or Fisher’s exact tests. Time-lapse data were not normally distributed; hence we used the nonparametric Mann–Whitney *U* test. We analyzed the re-expansion speed and multiple variables (embryo quality and ongoing pregnancy) using multivariate logistic regression to predict the pregnancy outcomes. The area under the curve was calculated using the receiver operating characteristic (ROC) curve, and a cutoff value was obtained. A *p*-value < 0.05 was considered statistically significant.

## 3. Results

### 3.1. General Characteristics of the Patients

The general characteristics of the patients for SVBTs and previous fresh IVF cycles are shown in Table 1. In total, 658 vitrified-warmed blastocysts that had undergone SVBTs were included in the study. The diameter of re-expansion of these vitrified-warmed blastocysts varied from 115 to 220 μm with a median of 145 μm and a mean of 144 ±14.3 μm. In ~56% (*n* = 368) of the vitrified-warmed blastocysts, the diameter varied between 135 and 156 μm; therefore, we used this value as a cutoff to classify the vitrified-warmed blastocysts. Accordingly, the 658 vitrified-warmed blastocysts were separated into four groups: Group I, <135 μm (*n* = 158); Group II, 135–145 μm (*n* = 197); Group III, 146–155 μm (*n* = 171); and Group IV, ≥156 μm (*n* = 132). In terms of the demographic characteristics of the patients, a significant difference was observed in the etiology of female infertility between the groups (Group I: 41.1%, Group II: 32.5%, Group III: 42.7%, and Group IV: 50.8%; *p* < 0.01). According to the previous fresh cycles of the patients, significant differences were observed in the fertilization method, total vitrification, and day of vitrification between Group IV and the other three groups; however, no significant difference was observed in the other parameters.

### 3.2. Laboratory and Clinical Outcomes

#### 3.2.1. Phase 1: Diameter

The time interval between the warming and transfer of an embryo was 16–28 h. Blastocyst diameter increased as the time interval was extended, and the degree of expansion was highly dependent on the diameter. However, as an exception, Group I showed a relatively large proportion of a fully or partially expanded blastocyst stage (Figure 1a). The blastocyst diameter in Groups II, III, and IV appeared to be good indicators of its quality, but small-sized blastocysts in Group I also exhibited good quality, though less frequently (Figure 1b). The clinical outcomes of the SVBT cycle followed by blastocyst diameter measurement did not show a significant difference; however, a tendency for a higher rate of positive β-hCG test results, clinical pregnancy, and ongoing pregnancy was observed (Figure 1c).

#### 3.2.2. Phase 2: Re-Expansion Speed

Based on blastocyst re-expansion speed distribution patterns, a strong correlation between re-expansion speed and ongoing pregnancy was observed (Figure 2a). Figure 2b shows the relationship between dynamic morphokinetic annotation and blastocyst quality, wherein the faster re-expansion speed resulted in high blastocyst quality (*p* < 0.001). Figure 2c shows the relationship between blastocyst diameter (µm; *x*-axis) and re-expansion speed (μm/min; *y*-axis), which displays a tendency toward direct proportionality. Subsequently, we compared the ongoing pregnancy outcomes of the embryo transfers to blastocyst diameter and re-expansion speed. Although no significant difference was observed in Group I, a tendency of positive correlation in Groups II, III, and IV was observed (Figure 2d).

### 3.3. Multivariate Logistic Regression Analysis of Blastocyst Quality

Multivariate logistic regression analysis was performed to predict the factors affecting blastocyst quality (Table 2). Embryo diameter, re-expansion speed, and retrieved number of oocytes were significantly associated with blastocyst quality. Both univariate and multivariate logistic regression analysis revealed that the blastocyst quality was significantly associated with the blastocyst diameter, speed of blastocyst re-expansion, and number of retrieved oocytes (Table 2). However, shorter hatching start points showed significant association only in univariate logistic regression analysis (*p* = 0.000). Maternal age at oocyte retrieval (years) and the number of fertilized embryos did not show an association with blastocyst quality.

### 3.4. Multivariate Logistic Regression Analysis of Ongoing Pregnancy

We also assessed the independent variables related to the ongoing pregnancy rate (Table 3). According to the univariate logistic regression analysis, maternal age (<38 year, OR = 2.170, 95% CI [1.414–3.331], *p* = 0.001), day of vitrification (Day 5, OR = 1.846, 95% CI [1.146–2.971], *p* = 0.012), and blastocyst quality (good, OR = 0.543, 95% CI [0.359–0.822], *p* = 0.004), re-expansion speed (≥50.1–≤100 μm/min, OR = 1.633, 95% CI [1.082–2.464], *p* = 0.02; ≥100.1 μm/min, OR = 2.096, 95% CI [1.356–3.240], *p* = 0.001) were correlated with ongoing pregnancy. In contrast, infertility duration and endometrial thickness demonstrated no effect on it. Multivariate logistic regression analysis revealed a significant difference between maternal age (<38 years, OR = 1.939, 95% CI [1.245–3.014], *p* = 0.003), day of vitrification (Day 5, OR = 2.027, 95% CI [1.237–3.311], *p* = 0.005), embryo quality score (good, OR = 0.641, 95% CI [0.411–0.999], *p* = 0.05), re-expansion speed (≥100.1 μm/min, OR = 1.931, 95% CI [1.159–2.994], *p* = 0.01), and ongoing pregnancies.

### 3.5. Clinical and Ongoing Pregnancy Based on Quantitative (Speed) and Qualitative (BQS) Selection Criteria

After ROC analysis, we looked for a BQS cutoff value and re-expansion speed with similar sensitivity and specificity. The ROC curve determined 59.8 μm/min as the optimal cutoff value for re-expansion speed and BQS to be 7 (Table 4). Transfer of single blastocysts with BQS 7 and a re-expansion speed of 59.8 μm/min resulted in a significantly higher rate of clinical pregnancy (62.4%) than the transfer of the blastocysts that did not meet these criteria (32.6%). Similarly, the ongoing pregnancy rate was significantly higher when using the blastocysts that met the cutoff values than that achieved using blastocysts that did not meet these criteria (55.9% vs. 27.4%; *p* < 0.01; Table 5). These results indicate that a combination of quantitative (speed) and qualitative (BQS) approaches is ideal for selecting competent embryos and predicting successful pregnancy.

## 4. Discussion

Selecting embryos with high developmental competence is a major issue in ART [1,4]. Research interests in identifying markers for embryo selection have increased to reduce the risks associated with multiple pregnancies and increase the chances of successful IVF procedures by adopting single-embryo transfer [4,7]. Therefore, an accurate evaluation of the morphological and morphokinetic parameters that drive the highest implantation potential is crucial for clinical outcomes. In accordance, the Korea National Health Insurance Law has set guidelines for the number of embryos to be transferred as a quality care parameter in ARTs.

In this study, we investigated the parameters affecting SVBT and its vitrification-warming process, specifically blastocyst diameter and re-expansion speed. We used a TLS to investigate all vitrification-warming cycles and found that the SVBT diameter and re-expansion speed correlated with the clinical outcomes. Our results show that quantitative assessment of the blastocyst stage could be useful for embryo selection. Moreover, embryo survival depends on quality grading and must meet the freezing threshold to be eligible for cryopreservation. Our results confirmed a high reliability of vitrification with a 98.35% overall survival rate after warming, substantiating that the vitrification protocol used here is efficient and does not negatively impact the embryo survival rate.

Several studies have highlighted the importance of the association between blastocyst diameter and clinical outcomes [19,20,31]. Huang et al. showed a significantly higher number of euploid blastocysts during the most rapid blastocoel expansion using 188 autologous blastocysts that underwent pre-implantation genetic testing for aneuploidies (PGT-A) [20]. Sciorio et al. suggested that transferring a larger blastocyst (diameter: 184 μm) achieved clinical pregnancy compared to that achieved by transferring smaller blastocysts (160 μm) [31]. Marion et al. assessed the association between blastocyst measurements and ongoing pregnancy and fertilization methods. Although the blastocoel of larger and faster embryos resulted in higher ongoing pregnancy rates, the dynamics of its expansion varied with the fertilization method; its size was higher in IVF embryos than in those of intracytoplasmic sperm injection (ICSI) or testicular sperm extraction (TESE-ICSI) [25]. However, other studies have reported contradicting results about the relationship between blastocyst diameter and clinical outcomes [22,23]. Almagor et al. reported that the inner cell mass diameter-to-blastocyst ratio rather than trophectoderm grade predicts successful clinical outcomes [23]. Lagalla et al. showed that a more expanded blastocyst resulted in a significantly higher implantation rate, but the mean area of blastocysts did not affect the implantation rate [22]. Kyoya et al. also suggested that reaching an experimentally determined blastocyst diameter (≥170 μm) did not affect the clinical outcomes. Pregnancy rates following frozen–thawed embryo transfer were constant irrespective of the blastocyst diameter [32]. Coello et al. reported that small-diameter blastocysts were less likely to plant, which provides reasonable and practical value in the laboratory [24,33]. Together with these studies, our study suggests that small blastocysts showed a tendency of increased blastocoel volume during the recovery period after warming. Our findings demonstrate a high proportion of fair- and poor-grade embryos and a relatively low proportion of good-quality embryos in Group I blastocysts (<135 µm diameter). Although we did not observe a significant difference in clinical outcomes, a tendency for increased ongoing pregnancy rate with increased blastocyst diameter was observed. Therefore, we suggest that a blastocyst diameter ≥135 µm is an effective indicator for cryopreservation.

The importance of blastocyst re-expansion can diverge across various spectra. Traditionally, many studies have mentioned the importance of blastocyst re-expansion [33,34,35]. However, recent studies have shown that embryo re-expansion does not affect clinical outcomes [36,37]. Cruz et al. also demonstrated that immediate embryo transfer after warming seemingly did not affect implantation potential [38]. Upon close examination of the timing of blastocoel re-expansion, Lin et al. showed that blastocysts with a faster re-expansion time (<1 h) resulted in a higher clinical pregnancy rate than those with a slow re-expansion time (>2 h) [39]. In contrast, Coello et al. found no difference between blastocysts that started re-expansion immediately and those that started it within 2 h post-warming [33]. We used digital imaging to analyze the blastocyst re-expansion speed by measuring the total migration length that reached the zona pellucida with the total time of filled re-expansion appearance. This corroborates our new approach to analyzing blastocyst re-expansion. Human embryos undergo continuous cell division to form blastocysts [40,41]. Dynamic change in speed is a potential indicator of developmental potential because the embryo changes prominently within a short period [42,43]. Most significantly, our previous study showed that morphological characteristics are the most important indicators of clinical outcomes and live birth [30,44]. Concordantly, in this study, we demonstrated that blastocyst diameter and re-expansion speed are closely related to embryo quality. Furthermore, we also showed that re-expansion speed was highly significantly correlated with ongoing pregnancy. Therefore, we suggest that blastocyst re-expansion speed is correlated with embryonic developmental potential. According to the multivariate logistic regression analysis, re-expansion speed is highly correlated with ongoing pregnancy. Therefore, the blastocyst re-expansion speed is correlated with embryonic developmental potential. Our results confirm those of previous studies showing that small diameter and slow speed or both were associated with poor embryo quality and cryo-survival rates, and consequently poor ongoing pregnancy rate per SVBT (≤31%). In agreement with previous studies [45,46], our findings also indicate that the decision to proceed with SVBT in the case of small and slow blastocysts may be uncertain.

Successful pregnancy can be influenced by the interaction between uterine receptivity and blastocyst quality [47,48]. Maternal age and embryo quality are major issues regarding ongoing pregnancy [46], but other factors such as uterine receptivity, age, freezing day, and blastocyst features can also affect the clinical results [15,49,50]. A previous study reported that blastocyst development highly depends on female age and blastocyst diameter [19,46]. Furthermore, Marion et al. showed a correlation between larger blastocoel size and faster expansion speed in relation to ongoing pregnancy [25]. Concordant with these studies, our study showed a strong association between maternal age at ET (≤38 years) and the rate of ongoing pregnancy. In addition, the blastocyst re-expansion speed decreased with increasing maternal age (Figure 3).

Our study provides several advantages for laboratory technicians and clinicians. To our knowledge, this is the first report of digital imaging using a combination of quantitative and qualitative embryo analysis to improve embryo selection. The strengths of our study are as follows. First, this was a single-center exhaustive study that included patients with various prognoses and heterogeneous baseline characteristics. Second, all IVF cycles were performed under uniform conditions, and blastocyst measurements were performed by a proficient embryologist to eliminate observer bias. Finally, post-warming parameters can be useful while counseling patients regarding their blastocyst quality and determining patient-oriented strategies. Such research strategies provide insights into clinical usability by assessing embryo potential.

Our study also had several limitations. First, this was a retrospective study from a single fertility center; the observations should be confirmed in multi-center studies. Second, we found that Group IV with a larger blastocyst (size) had significantly few embryos produced by the ICSI method, and significantly more embryos were vitrified on D6. These two parameters could have contributed to influencing the result. However, the fertilization methods were not significantly correlated with embryo quality and ongoing pregnancy outcomes. It is necessary to examine other parameters to validate our findings.

It will be interesting to identify the underlying reason for slow-developing embryos and consider the compaction state of the morula (partial compaction vs. full compaction), which may play a crucial role in blastocyst development. Fine-tuned blastocyst selection based on maternal age, diameter, and speed also needs to be considered in future research. Larger prospective randomized trials are required to validate their clinical relevance.

## 5. Conclusions

In summary, improving the efficacy of ART programs has always been an important issue. We have closely examined blastocyst diameter and speed. Both of these parameters are important from the embryo quality perspective, whereas age, freezing day, and embryo quality are crucial from a clinical perspective (Figure 4). Our combined quantitative (diameter and speed) and qualitative (BQS) approach can help in evaluating the clinical usability of embryos.

## Figures and Tables

**Figure 1 jcm-11-07085-f001:**
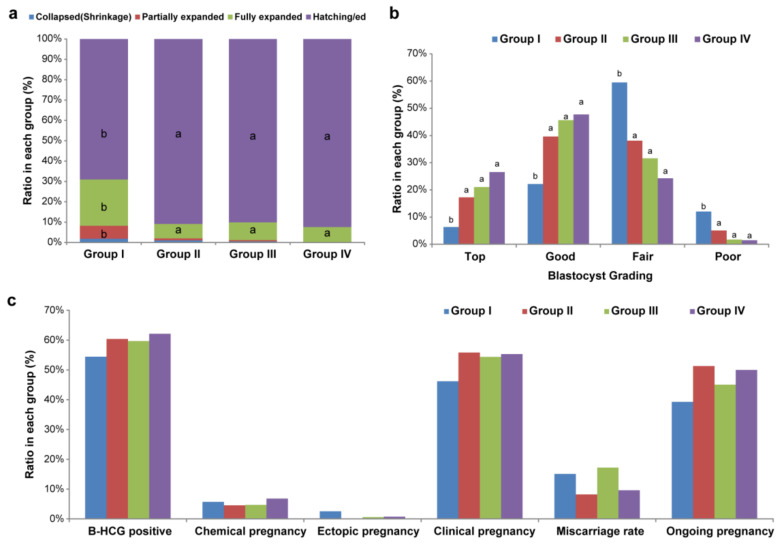
Clinical results of single vitrified-warmed transfer (SVBT) cycles (distribution patterns of blastocyst diameter). (**a**) Degree of re-expansion according to blastocyst diameter during the interval between warming and transfer. (**b**) Different morphologic grading of blastocyst single vitrified-warmed transfer cycles. (**c**) Comparison of clinical outcome of embryo transfer categories according to blastocyst diameter size. ^a,b^ Different superscript letters denote significant differences (*p* < 0.001).

**Figure 2 jcm-11-07085-f002:**
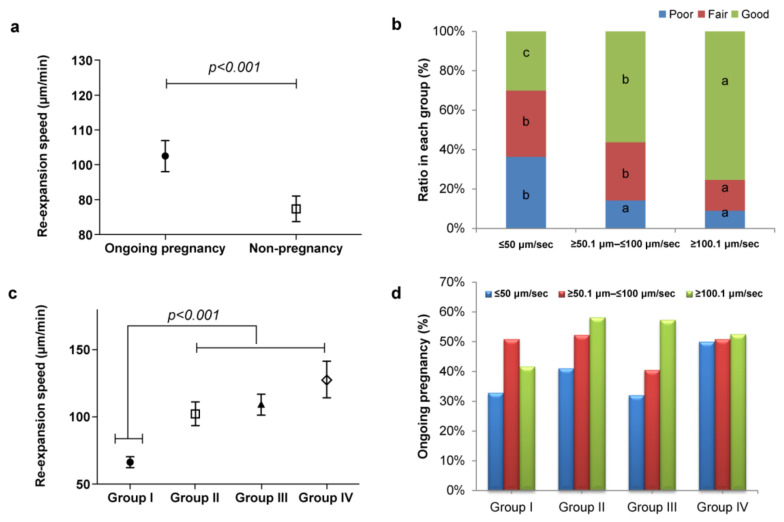
Distribution patterns of blastocyst re-expansion speed. (**a**) Distribution of re-expansion speed values between ongoing pregnancy and non-pregnancy. (**b**) Relationship between dynamic morphokinetic (re-expansion speed) annotation and blastocyst quality *p* < 0.001. (**c**) Relationship between blastocyst diameter (*x*-axis) and re-expansion speed (μm/min; *y*-axis). (**d**) Ongoing pregnancy outcome of embryo transfers according to blastocyst diameter size and re-expansion speed. ^a,b,c^ Different superscript letters denote a significant difference (*p* < 0.001).

**Figure 3 jcm-11-07085-f003:**
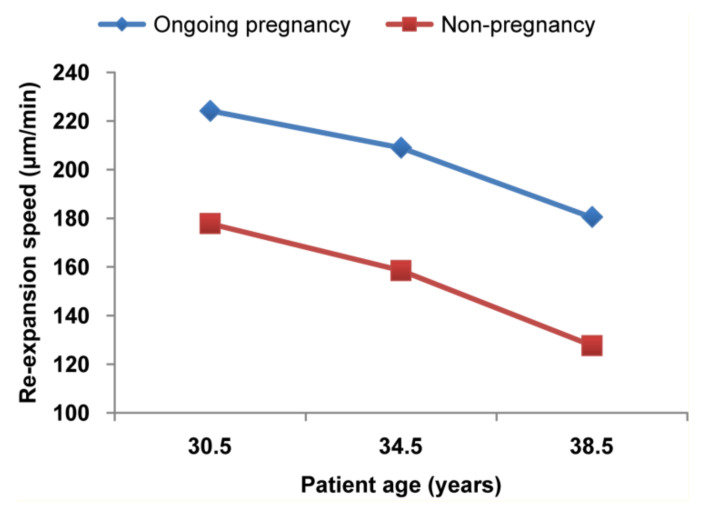
The correlation between blastocyst re-expansion speed and female age.

**Figure 4 jcm-11-07085-f004:**
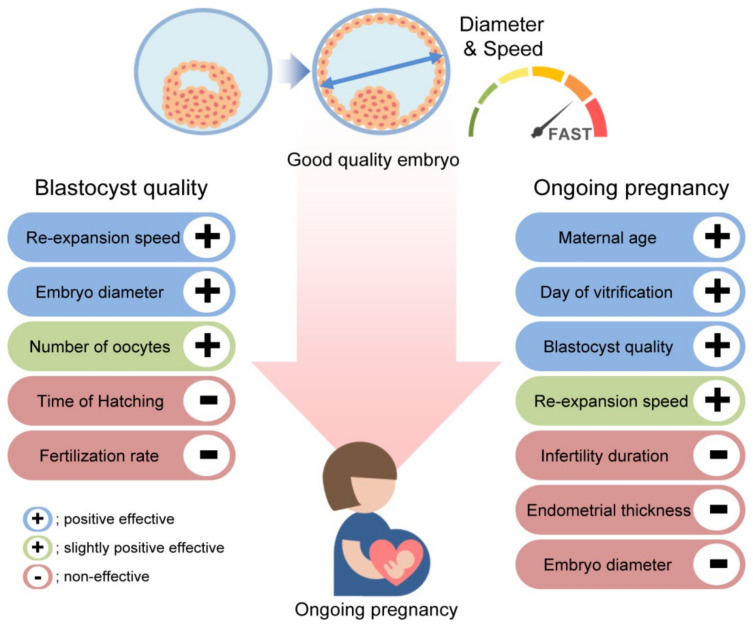
Scheme illustrating the variables identified in this study that can be used to achieve good-quality embryos and successful ongoing pregnancy. An increase in diameter and re-expansion speed positively affects embryo quality, whereas maternal age at oocyte retrieval, day of vitrification, and quality of embryos are correlated with ongoing pregnancy.

**Table 1 jcm-11-07085-t001:** Basic clinical data of patients in this study.

Characteristics	Variables	All Transfer	Group I	Group II	Group III	Group IV	*p*-Value
Demographic characteristics	Number of FET cycles	658	158	197	171	132	
	Cryo survival rate	98.35 ± 0.09	96.60 ± 0.12	98.89 ± 0.08	98.93 ± 0.07	98.86 ± 0.07	0.24
	Mean maternal age (years, ±SD)	33.93 ± 3.59	33.97 ± 4.13	33.67 ± 3.38	34.14 ± 3.63	33.97 ± 3.16	0.65
	Mean paternal age (years, ±SD)	36.98 ± 4.54	37.53 ± 5.69	36.90 ± 4.28	37.01 ± 4.06	36.44 ± 3.93	0.25
	Etiology of infertility [(%) n]						
	Female	40.9% (269)	41.1% (65) ^a^	32.5% (64) ^a,b^	42.7% (73) ^a^	50.8% (67) ^a,c^	0.01
	Male	19.0% (125)	13.9% (22)	24.4% (48)	21.1% (36)	14.4% (19)	0.08
	Combine (male + female)	25.5% (168)	30.4% (48)	29.4% (58)	21.1% (36)	19.7% (26)	0.06
	Unexplained	12.2% (80)	10.1% (16)	11.2% (22)	13.5% (23)	14.4% (19)	0.64
	Other	2.4% (16)	4.4% (7)	2.5% (5)	1.8% (3)	0.8% (1)	0.21
	EM thickness at hCG (mm)	9.59 ± 1.70	9.61 ± 1.82	9.71 ± 1.76	9.51 ± 1.56	9.48 ± 1.66	0.58
	FET cycle protocol						
	Natural [(%) *n*]	81.6% (537)	79.7% (126)	85.3% (168)	80.1% (1370	80.3% (106)	0.47
	HRT [(%) *n*]	18.4% (121)	20.3% (32)	14.7% (29)	19.9% (34)	19.7% (26)	0.35
Characteristics of previous IVF cycle	Maternal age at oocyte retrieval (years, ±SD)	33.52 ± 3.61	33.97 ± 4.13	33.13 ± 3.37	33.62 ± 3.66	33.48 ± 3.16	0.18
	Paternal age at oocyte retrieval (years, ±SD)	36.49 ± 4.42	37.24 ± 5.36	36.32 ± 4.21	36.41 ± 4.03	36.00 ± 3.92	0.10
	Infertility duration (y)	3.19 ± 2.2	3.14 ± 2.27	3.07 ± 2.11	3.4 ± 2.34	3.15 ± 2.07	0.62
	Number of previous IVF attempts	1.54 ± 1.02	1.63 ± 1.30	1.52 ± 0.94	1.62 ± 1.01	1.35 ± 0.74	0.09
	BMI (kg/m^2^)	21.13 ± 2.65	21.35 ± 2.77	20.98 ± 2.51	21.23 ± 2.80	20.97 ± 2.53	0.52
	Parity	0.10 ± 0.33	0.09 ± 0.29	0.09 ± 0.36	0.12 ± 0.35	0.08 ± 0.30	0.70
	AMH (ng/mL)	4.26 ± 3.47	4.26 ± 3.77	4.51 ± 3.53	4.33 ± 3.70	3.81 ± 2.62	0.37
	Basal FSH (IU/L)	7.19 ± 2.97	7.54 ± 3.25	7.16 ± 3.35	6.85 ± 2.54	7.26 ± 2.48	0.23
	Basal LH (IU/L)	5.97 ± 3.60	5.81 ± 3.16	5.97 ± 3.62	5.88 ± 4.07	6.26 ± 3.46	0.76
	Basal E2 (IU/L)	44.32 ± 22.19	44.49 ± 18.51	43.88 ± 17.61	45.93 ± 32.24	42.72 ± 15.98	0.68
	Basal TSH (mIU/mL)	1.56 ± 1.24	1.42 ± 0.81	1.62 ± 1.32	1.60 ± 1.39	1.58 ± 1.33	0.47
	Basal Prolactin (ng/mL)	15.51 ±10.04 10.0401	15.69 ± 11.826	15.01 ± 9.40	15.965 ± 10.133	15.48 ± 8.51	0.84
	MCD-AFC	16.66 ± 9.66	15.33 ± 10.63	17.81 ± 8.52	17.20 ± 10.9	15.84 ± 8.21	0.09
	Ovarian stimulation protocol						
	Antagonist [(%) *n*]	93.0% (612)	93.7% (148)	93.4% (184)	90.6% (155)	94.7% (125)	0.53
	Agonist [(%) *n*]	3.2% (21)	3.2% (5)	3.6% (7)	4.1% (7)	1.5% (2)	0.63
	Natural [(%) *n*]	3.8% (25)	3.2% (5)	3.0% (6)	5.3% (9)	3.8% (5)	0.69
	FSH total dose	1503.9 ± 469.94	1521.1 ± 475.66	1458.1 ± 439.13	1518.6 ± 484.49	1533.8 ± 489.93	0.48
	Fertilization method						
	Standard IVF [(%) *n*]	31.3% (206)	26.6% (42) ^a^	22.8% (45) ^a^	31.0% (53) ^a^	50.0% (66) ^b^	0.01
	ICSI [(%) *n*]	60.2% (396)	63.9% (101) ^a^	71.1% (140) ^a^	60.8% (104) ^a^	38.6% (51) ^b^	0.01
	Half ICSI [(%) *n*]	8.5% (56)	9.5% (15)	6.1% (12)	8.2% (14)	11.4% (15)	0.38
	Vitrification total						
	Surplus [(%) *n*]	25.8% (170)	17.1% (27) ^a^	26.4% (52) ^a^	25.1% (43) ^a^	36.4% (48) ^a,b^	0.01
	All freezing [(%) *n*]	74.2% (488)	82.9% (131) ^a^	73.6% (145) ^a^	74.9% (128) ^a^	63.6% (84) ^a,b^	0.01
	Mean number of retrieved oocytes (*n*)	16.64 ± 9.04	16.44 ± 9.61	16.95 ± 9.12	16.82 ± 9.15	16.19 ± 8.12	0.88
	Fertilization embryos (2 pronuclei)	11.05 ± 6.1	10.71 ± 6.59	11.03 ± 5.72	11.2 ± 6.19	11.3 ± 5.98	0.85
	Mean number of frozen blastocysts (*n*)	3.32 ± 2.4	3.23 ± 2.32	3.26 ± 2.31	3.35 ± 2.44	3.44 ± 2.55	0.89
	Day of vitrification [(%) *n*]						
	D5	86.9% (572)	92.4% (146) ^a^	92.4% (182) ^a^	88.9% (152) ^a^	69.7% (92) ^b^	0.01
	D6	13.1% (86)	7.6% (12) ^a^	7.6% (15) ^a^	11.1% (19) ^a^	30.3% (40) ^b^	0.01

Note: Group I; <135 μm, Group II; ≥135–≤145 μm, Group III; ≥146–≤155 μm, Group IV; ≥156 μm. Values are presented as the mean ± standard deviation (SD). EM, endometrial; FET, frozen embryo transfer; BMI, body mass index; AMH, anti-Mullerian hormone; FSH, follicle-stimulating hormone; LH, luteinizing hormone; E2, estradiol hormone; TSH, thyroid-stimulating hormone; MCD, menstrual cycle day; AFC, antral follicle count; HRT, hormone replacement treatment. ^a,b,c^ Different superscript denotes a significant difference (*p* < 0.05).

**Table 2 jcm-11-07085-t002:** Results of the univariate and multivariate logistic regression analyses to predict the blastocyst quality.

Variables	OR (95% CI)	*p*-Value	Adjusted OR (95% CI)	*p*-Value
Maternal age at oocyte retrieval (years)				
≤30	1.070 (0.609–1.879)	0.814	n/a	
31–34	1.252 (0.802–1.955)	0.322	n/a	
≥35	Reference			
Embryo diameter (μm)				
<135	Reference		Reference	
≥135–≤145	4.801 (2.888–7.982)	0.000	2.815 (1.567–5.056) *	0.001
≥146–≤155	9.668 (5.063–18.460)	0.000	5.315 (2.584–10.934) *	0.000
≥156	12.330 (5.647–26.919)	0.000	7.917 (3.113–20.135) *	0.000
Embryo re-expansion speed (μm/min)				
≤50	Reference		Reference	
≥50.1–≤100	3.443 (2.141–5.534)	0.000	1.774 (0.994–3.167) *	0.053
≥100.1–≤150	5.774 (3.228–10.330)	0.000	2.484 (1.232–5.008) *	0.011
tHSP (hatching start point) (h)				
≤4	2.959 (1.910–4.586)	0.000	n/a	
≥4.1	Reference			
Number of retrieved oocytes				
≤10	Reference		Reference	
≥11–≤20	1.823 (1.108–2.999)	0.018	2.019 (1.122–3.632) *	0.019
≥21	1.311 (0.789–2.179)	0.295	n/a	
Number of fertilization embryos				
≤7	Reference			
≥ 8–≤16	1.465 (0.936–2.294)	0.950	n/a	
≥17	1.810 (0.951–3.444)	0.071	n/a	

* Adjusted ORs are reported for variables in the final, n/a, not applicable.

**Table 3 jcm-11-07085-t003:** Results of the univariate and multivariate logistic regression analyses to predict the ongoing pregnancy outcome.

Variables	OR (95% CI)	*p*-Value	Adjusted OR (95% CI)	*p*-Value
Maternal age at ET (years)				
<38	2.170 (1.414–3.331)	0.001	1.939 (1.245–3.014) *	0.003
≥38	Reference		Reference	
Infertility duration (years)				
≤2	1.152 (0.077–1.722)	0.490	n/a	
≥3–≤4	1.013 (0.656–1.565)	0.953	n/a	
≥5	Reference		n/a	
Endometrial thickness in VBT (mm)				
<10	0.766 (0.563–1.043)	0.091	n/a	
≥10	Reference		n/a	
Day of vitrification				
Day 5	1.846 (1.146–2.971)	0.012	2.027 (1.237–3.311) *	0.005
Day 6	Reference		Reference	
Blastocyst quality score				
good	0.543 (0.359–0.822)	0.004	0.641 (0.411–0.999) *	0.050
poor	Reference		Reference	
Embryo diameter (μm)				
<135	Reference		n/a	
≥135–≤145	1.629 (1.066–2.490)	0.024	n/a	
≥146–≤155	1.268 (0.818–1.967)	0.289	n/a	
≥156	1.548 (0.970–2.471)	0.067	n/a	
Embryo re-expansion speed (μm/min)				
≤50	Reference		Reference	
≥50.1–≤100	1.633 (1.082-2.464)	0.020	n/a	
≥100.1	2.096 (1.356-3.240)	0.001	1.931 (1.159–2.994) *	0.010

* Adjusted ORs are reported for variables in the final logistic regression model after backward stepwise selection; n/a, not applicable.

**Table 4 jcm-11-07085-t004:** The optimal cutoff values of the ongoing pregnancy for the determination of re-expansion speed and BQS.

	AUROC	95% CI	Optimal Cutoff Value	Sensitivity	Specificity	PPV	NPV	*p*-Value
Speed	0.599	0.556–0.641	59.8	0.75	0.38	0.53	0.37	<0.0001
BQS	0.592	0.549–0.635	7	0.78	0.39	0.53	0.33	<0.0001

Note: The optimal cutoff value is the one that gives the higher total sensitivity and specificity. Speed, re-expansion speed; BQS, blastocyst quality scoring; AUROC, areas under the receiver operator characteristic curves; PPV, positive predicted value; NPV, negative predicted value.

**Table 5 jcm-11-07085-t005:** Clinical pregnancy and ongoing pregnancy characteristics based on quantitative (speed) and qualitative (BQS) selection criteria: BQS of 7 and re-expansion speed of 59.8 μm/min.

Time-Lapse Selection Criteria	Applicable	Not Applicable	*p*-Value
No. of single blastocysts transferred	340	95	
No. of clinical pregnancy [*n* (%)]	212 (62.40%)	31 (32.60%)	<0.0001
No. of abortions [*n* (%)]	22 (6.50%)	5 (5.30%)	NS
No. of ongoing pregnancy [*n* (%)]	190 (55.90%)	26 (27.40%)	<0.0001

NS, not significant.

## Data Availability

Not applicable.

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
