# Peer review of "Clinical Usability of Embryo Development Using a Combined Qualitative and Quantitative Approach in a Single Vitrified-Warmed Blastocyst Transfer: Assessment of Pre-Vitrified Blastocyst Diameter and Post-Warmed Blastocyst Re-Expansion Speed"

_jcm, 2022, doi:10.3390/jcm11237085_

Round 1

Reviewer 1 Report

Park et al. carried out a quantitative analysis on more than 600 vitrified human blastocysts, trying to correlate the cyst diameters and re-expansion speeds of the vitrified-thaw blastocyst with pregnancy outcomes. This is a one-center based study, but overall, the study is well-designed. Though the study observation should be confirmed in other center, the results obtained may be useful for the embryologist to select the best blastocyst for single embryo transfer. I suggest the publication of the paper after the following questions have been addressed.

 1.         How were the grouping of blastocyst size determined? Any reason why <135 μm, ≥135–≤145 μm (n = 197); Group III, ≥146– 77, ≤155 μm (n = 171); and Group IV, ≥156 μm (n = 132). Similarly, how did they define the blastocyst quality “Blastocyst quality was divided into four groups based on their morphological scores: top (blastocyst score: 45–24), good (23–16), average (15–7), and poor (6–1).”?

2.         The group with larger blastocyst (size) has significantly less embryos produced by ICSI method and significantly more embryos were vitrified on D6. These two parameters could have contributed to the current difference found such as blastocyst grading and pregnancy outcomes, and should be included in table 2 (both) and table 3 (ICSI). Appropriate discussion on these should be included as well.

Author Response

첨부파일을 참조하세요

Reviewer 2 Report

Good evening and congratulations on your work!

The study is well conducted and the large lot of cases included together with the clear and thorough analysis make up for a good quality paper. The study of embryo quantitative (diameter and speed) and qualitative scores can benefit both the lab workers and the clinicians, resulting in higher pregnancy rates in IVF patients. There have been many studies that mention the importance of blastocyst reexpanstion throughout literature, many of which you have mentioned in the article, showing good documentation. 

Author Response

Dear Reviewer, thank you for your favorable feedback and recommendation. We deeply appreciate your time and efforts in reviewing our manuscript.
